# Antimetastatic Effects of Sesamin on Human Head and Neck Squamous Cell Carcinoma through Regulation of Matrix Metalloproteinase-2

**DOI:** 10.3390/molecules25092248

**Published:** 2020-05-10

**Authors:** Jian-Ming Chen, Pei-Yin Chen, Chia-Chieh Lin, Ming-Chang Hsieh, Jen-Tsun Lin

**Affiliations:** 1Department of Surgery, Kaohsiung Armed Forces General Hospital, Kaohsiung 80284, Taiwan; cvsming@gmail.com; 2Department of Recreation and Holistic Wellness, MingDao University, Changhua 523, Taiwan; payin0923@mdu.edu.tw; 3Oral Cancer Research Center, Changhua Christian Hospital, Changhua 500, Taiwan; 181327@cch.org.tw; 4School of Medical Laboratory and Biotechnology, Chung Shan Medical University, Taichung 40201, Taiwan; 5Department of Clinical Laboratory, Chung Shan Medical University Hospital, Taichung 40201, Taiwan; 6Division of Hematology and Oncology, Department of Medicine, Changhua Christian Hospital, Changhua 500, Taiwan; 7School of Medicine, Chung Shan Medical University, Taichung 402, Taiwan

**Keywords:** sesamin, human head and neck squamous carcinoma, MMP-2, migration, invasion, MAPK

## Abstract

Background: Sesamin is a lignin present in sesame oil from the bark of *Zanthoxylum* spp. Sesamin reportedly has anticarcinogenic potential and exerts anti-inflammatory effects on several tumors. Hypothesis/Purpose: However, the effect of sesamin on metastatic progression in human head and neck squamous carcinoma (HNSCC) remains unknown in vitro and in vivo; hence, we investigated the effect of sesamin on HNSCC cells in vitro. Methods and Results: Sesamin-treated human oral cancer cell lines FaDu, HSC-3, and Ca9-22 were subjected to a wound-healing assay. Furthermore, Western blotting was performed to assess the effect of sesamin on the expression levels of matrix metalloproteinase (MMP)-2 and proteins of the MAPK signaling pathway, including p-ERK1/2, P-p38, and p-JNK1/2. In addition, we investigated the association between MMP-2 expression and the MAPK pathway in sesamin-treated oral cancer cells. Sesamin inhibited cell migration and invasion in FaDu, Ca9-22, and HSC-3 cells and suppressed MMP-2 at noncytotoxic concentrations (0 to 40 μM). Furthermore, sesamin significantly reduced p38 MAPK and JNK phosphorylation in a dose-dependent manner in FaDu and HSC-3 cells. Conclusions: These results indicate that sesamin suppresses the migration and invasion of HNSCC cells by regulating MMP-2 and is thus a potential antimetastatic agent for treating HNSCC.

## 1. Introduction

Cancer-related morbidity and mortality rates have increased worldwide, and the causes of cancer remain complex. In total, there were 18.1 million new cancer patients and 9.6 million people were predicted to die of cancer in 2018 [1]. Among these individuals, approximately 40% were predicted to be diagnosed with head and neck cancer, of which approximately 170,000 were predicted to die as a consequence [1]. Head and neck squamous cell carcinoma (HNSCC) may manifest in the nasopharynx, nasal and paranasal sinuses, buccal cavity, oropharyngeal region, hypopharynx, larynx, salivary gland, or thyroid gland. Infection with human papillomaviruses (HPVs) has been implicated in the pathogenesis of HNSCC. HPV (+) and HPV (−) HNSCC have different molecular mechanisms in the oncogenic processes [2]. Over 90% of head and neck cancers are oral squamous cell carcinoma (OSCC) [3]. The primary treatments for head and neck cancer include surgical resection, radiation therapy, chemotherapy, molecular-targeted therapy, or a combination of these [4]. Notwithstanding numerous treatment modalities for oral cancer, the five-year survival rate remains poor because of its high metastatic potential [5]. Some evidence has revealed that reducing subjection to certain environmental and personal risk factors, such as by abstaining from smoking and drinking alcohol and taking measures to maintain a weight below the overweight threshold, can help to prevent oral cancer [6]. Nevertheless, studies continually aim to identify effective drugs and treatments for cancer.

Metastasis is usually one of the causes of high mortality among patients with head and neck cancer [7]. This is a complex phenomenon involving numerous enzymes. Matrix metalloproteinases (MMPs) are extensively associated with metastasis, tumorigenesis, cardiovascular disease, and autoimmune diseases [8,9,10,11]. MMPs are generally required for normal physiological function, and their aberrant expression may result in pathological conditions. For instance, MMP-9 and MMP-2 upregulation is reportedly associated with cancer cell migration in various cancers [12,13]. Hsieh et al. reported that pinostilbene hydrate suppresses cell migration and invasion in OSCC [14]. Some studies have reported that in patients with breast cancer, MMP-2 suppression is a potential marker for a distinctly favorable prognosis [15].

Sesame is widely cultivated in Asia and Africa and is considered a good nutritional supplement that is rich in fat, sesamin, and sesamolin. In particular, sesamin (SES) has multiple bioactive effects, including anti-inflammatory, antitumor, antioxidant, and apoptotic effects [16,17,18,19]. Deng et al. reported that sesamin inhibits IL-6-induced gp130/STAT3 signaling by mediating G2/M cell cycle arrest and inhibiting proliferation in the human hepatocellular carcinoma cell line HepG2 [20]. The tyrosine phosphorylation of EphA1 and EphB2 reportedly regulates the autophagy induced by sesamin in colon cancer cells [21]. In human embryonic kidney carcinoma A293 cells, the NF-κB-dependent reporter expression of TNF, TNFR1, TRADD, NIK, and IKK can be inhibited by sesamin [22]. Furthermore, sesamin can potentiate apoptosis and suppress human chronic myeloid leukemia cell proliferation [22]. Hence, sesamin is a potentially potent chemopreventive drug for colon and liver cancer; however, the mechanism underlying its antimetastatic effects in HNSCC remain unknown. Therefore, we investigate the mechanism underlying the antimetastatic effect of sesamin in HNSCC and on MMP signaling.

## 2. Materials and Methods

### 2.1. Cell Culture

The human tongue squamous carcinoma HSC-3 and pharynx squamous carcinoma FaDu cell lines were obtained from ATCC (Manassas, VA, USA). The human gingiva squamous carcinoma cell line Ca9-22 was purchased from and validated by the Japanese Collection of Research Bioresources Cell Bank. The Ca9-22 and FaDu cell lines were cultured in Eagle’s Minimum Essential Medium (MEM; Life Technologies, Grand Island, NY, USA) with fetal bovine serum to a final concentration of 10%. The HSC-3 was cultured in Dulbecco’s Modified Eagle Medium (DMEM; Life Technologies, Grand Island, NY, USA) supplemented with an equal volume of Ham’s F12 Nutrient Mixture (Life Technologies, Grand Island, NY, USA) and also supplemented with fetal bovine serum to a final concentration of 10%.

### 2.2. Sesamin Treatment

Sesamin is a pure substance bought from Sigma-Aldrich (St. Louis, MO, USA), the purity of which is over 98%. By using dimethyl sulfoxide (DMSO), sesamin stock was prepared at a concentration of 100 mM; respectively diluted to 0, 10, 20, and 40 mM; and stored at −20 °C. The content of DMSO in all medicines should be less than 0.2%. While processing when medicine was added, medicine at the appropriate concentration was mixed and added into the cell culture dish. Cells were put into an incubator at 37 °C and 5% CO_2_ following the required conditions and time for the experiment.

### 2.3. MTT (3-(4,5-Dimethylthiazol-2-yl)-2,5-diphenyltetrazolium Bromide) Assay

HSC-3, FaDu, and Ca9-22 cells were evenly distributed into 96-well plates with appropriate numbers of cells (1 × 10^4^ cells/well). After placement and cultivation in the 37 °C, 5% CO_2_ incubator for 24 h, used culture fluid was aspirated, and the cells were washed with 1x PBS, respectively adding culture fluid afterwards, which contained sesamin (0, 10, 20, or 40 μM), followed by incubation at 37 °C for 24 h. Later, 1x PBS was used to wash the cells and also removed in order to prevent the remaining medicaments and the function of MTT from interfering with the results. After washing twice, 0.8 mL/well of culture fluid containing MTT reagent (the ultimate MTT concentration was 0.5 mg/mL) was added and later removed after incubation in the 37 °C, 5% CO_2_ incubator for 4 h, and crystallized MTT was dissolved by 0.5~1 mL/well of isopropanol. The crystals were dissolved with methanol, and the absorbance was determined at 595 nm by utilizing the spectrophotometer (BioTek Synergy HTX Multi-Mode Reader, Winooski, VT, USA). Each concentration was repeatedly tested in three separate experiments.

### 2.4. Culture Inserts for the Wound Healing Assay

HSC-3, FaDu, and Ca9-22 oral cancer cells were evenly distributed into Culture-Inserts 2 Well for self-insertion on 6-well plates (2 × 10^5^ cells/well) and given different concentrations of sesamin (0, 10, 20, or 40 μM). After respectively progressing for different time periods, the conditions of wound healing were inspected under a microscope, and photos were taken as records of the images.

### 2.5. Cell Migration and Invasion Assays

HSC-3, FaDu, and Ca9-22 oral cancer cells were evenly distributed into 6 cm petri dishes with appropriate numbers of cells (4 × 10^5^ cells/well) and placed in a 37 °C, 5% CO_2_ incubator for 24-h cultivation. After removing the used culture fluid, sesamin (0, 10, 20, or 40 μM) was added for 37 °C, 24-h cultivation. By making use of the transwell inserts (1 × 10^4^ cells/well), 60 μL of culture fluid containing 100% fetal bovine serum was added to the lower layer, while a fixed quantity of 60 μL of culture fluid without the serum was added to the upper layer. After 24 h of cell movement, the cells were fixed for ten minutes by methanol, dried, and then dyed for four hours with 10× Giemsa, then non-moving cells on the upper layer of the film were removed. Three fields of vision were chosen per insert under a 100× microscope to count the numbers of cells. For the invasion assay, firstly, Matrigel was diluted ten times and spread on the upper coating, and secondly, the following steps from the migration assay were repeated.

### 2.6. Cell Protein Preparation

HSC-3, FaDu, and Ca9-22 oral cancer cells were evenly distributed into 6 cm petri dishes with appropriate numbers of cells (4 × 10^5^ cells/well) and placed in a 37 °C, 5% CO_2_ incubator for 24-h cultivation. After removing the used culture fluid, sesamin (0, 10, 20, or 40 μM) was added for 24-h cultivation, and RIPA buffer was added to collect the cells into 1.5 mL centrifuge tubes. The protein concentrations of total cell lysates were determined by the BCA protein assay. The absorbance was determined at 570 nm by utilizing the spectrophotometer (BioTek Synergy HTX Multi-Mode Reader). A fixed quantity of cell protein (20 μg) was taken after determining the concentration of the sample’s protein.

### 2.7. Western Blot Assay

The deployed protein samples were inserted into an SDS-PAGE gel and electrophoretically separated at 70 volts through the upper gel and 100 volts through the lower gel. After that, the proteins were transferred to the membrane at 120 volts at 4 °C for 1.5 h. The membrane was then blocked for an hour at room temperature with 5% non-fat milk. Afterwards, primary antibodies that were prepared in advance by dilution were added for incubation and shaking for 24 h at 4 °C after pouring away the blocking buffer, with unbound antibodies subsequently being washed away at room temperature with washing buffer 3 times, for 10 min each. Then, secondary antibodies that were prepared by dilution were added for incubation at room temperature for 1 h, with unbound antibodies subsequently being washed away. Finally, ECL was added, and signals were detected using the chemiluminescence fluorescence Image Quant LAS 4000 (GE Healthcare, Berlin, Germany) biomolecular imaging system. The relative photographic density photos were analyzed to quantify expression with the Image J software, and expression was normalized relative to β-actin.

### 2.8. Statistical Analysis

Statistical analyses were performed with Student’s t-test (in SigmaPlot 10.0, San Jose, CA, USA). The experiments were repeated at least 3 times and the values are indicated as mean ± SD. The significance level was set at *p* < 0.05.

## 3. Results

### 3.1. Cell Viability of Human Oral Cancer Cells after Sesamin Treatment

The chemical structure of sesamin is shown in Figure 1a. After the 24-h treatment of cells with 0, 10, 20, and 40 μM sesamin, cell viability and proliferation were assessed using an MTT assay, and we found that the viability of FaDu cells was not significantly affected (Figure 1b). The viability of the Ca9-22 and HSC-3 cells is depicted in Figure 1c,d. Although cell viability was affected by treatment with 40 μM sesamin, no significant cytotoxic effects were observed on either cell line. Thus, 0–40 μM sesamin was selected as the appropriate dose range for subsequent experiments.

### 3.2. Motility of Sesamin-Treated Human Oral Cancer Cells According to Results of a Wound-Healing Assay

To assess the coordinated movement of a cell population, wound-healing assays were performed for FaDu, HSC-3, and Ca9-22 cells with 0, 10, 20, and 40 μM sesamin. Figure 2a,b show the results of the wound-healing assay and quantitative analysis of FaDu cells. The cell motility of FaDu cells was markedly decreased upon 6-h treatment with 40 µM sesamin in comparison with the control group, as was that of HSC-3 cells after 6-h treatment with 40 µM sesamin (Figure 2c,d). Similar results were obtained for Ca9-22 cells after 24-h treatment with 40 µM sesamin (Figure 2e,f). These results indicate that sesamin inhibits the coordinated movement of the tumor cell population.

### 3.3. Invasion and Migration of Human Oral Cancer Cells after Sesamin Treatment

To assess the effect of sesamin on the invasion and migration of human oral cancer cells, FaDu, HSC-3, and Ca9-22 cells were subjected to a transwell assay. Sesamin inhibited cell migration in a dose-dependent manner in all three cell lines (Figure 3a). Cell migration was inhibited by >50% after treatment with 40 µM sesamin (Figure 3b). Furthermore, the invasiveness of the FaDu, HSC-3, and Ca9-22 cells markedly decreased upon treatment with 20 and 40 µM sesamin (Figure 3c,d). These results indicate that sesamin inhibits the invasion and migration of human oral cancer cells.

### 3.4. Effect of Sesamin on MMP-2 Expression in Human Oral Cancer Cells 

MMP-2 is associated with metastasis in many cancers [23]. To determine whether sesamin affects MMP-2 expression in oral cancer cells, HSC-3 and FaDu cells were subjected to Western blotting experiments. Figure 4a–c present the blots after treatment with 0, 10, 20, and 40 μM sesamin for 24 h. Densitometric analysis revealed that MMP-2 was downregulated by 23% and 19% in HSC-3 and FaDu cells, respectively, upon treatment with 40 µM sesamin (Figure 4b–d). These results indicate that sesamin downregulates MMP-2 in HSC-3 and FaDu cells.

### 3.5. Effect of Sesamin on the Association between MMP-2 and MAPK in Human Oral Cancer Cells 

The MAPK pathway is generally involved in cell migration [24]. Therefore, we assessed the expression of p-ERK1/2, p-p38, and p-JNK1/2 in FaDu and HSC-3 cells treated with sesamin through Western blotting. We found that P-p38 and p-JNK1/2 levels were markedly reduced after sesamin treatment at 40 µM in HSC-3 cells, whereas p-ERK1/2 levels were markedly increased (Figure 5a,b). Similar results were obtained for FaDu cells (Figure 5c,d). These results indicate that sesamin inhibits the migration of oral cancer cells by potentially downregulating P-p38 and p-JNK1/2 in the MAPK signaling pathway. On the basis of the aforementioned results, which indicated the downregulation of P-p38 and p-JNK1/2, we used SB203580 (a p38 inhibitor) and SP600125 (a JNK inhibitor) to confirm the association between MMP-2 expression and players in the MAPK signaling pathway in HSC-3 and FaDu cells. Firstly, cells were pretreated with SB203580 (20 μM) and SP600125 (20 μM) for 1 h. Thereafter, sesamin (40 μM) was supplemented in the culture medium, and cells were cultured for 24 h. The cell lysates were then collected for Western blotting. Combinatorial treatment with the p38 and JNK inhibitors and sesamin further downregulated MMP-2 in HSC-3 and FaDu cells in comparison with sesamin treatment alone (Figure 6a–c). Quantitative data are shown in Figure 6b–d. These results indicate that sesamin cooperatively downregulates MMP-2, p-JNK1/2, and P-p38 and inhibits the MAPK signaling pathway along with p38 and p-JNK inhibitors in oral cancer cells.

### 3.6. Effect of Combinatorial Treatment of MAPK Inhibitors and Sesamin on the Motility of Human Oral Cancer Cells

To determine the effect of the combined MAPK signaling inhibition and sesamin treatment on the motility of human oral cancer cells, HSC-3 and FaDu cells were pretreated with or without SB203580 (p38 inhibitor) and SP600125 (JNK inhibitor) for 1 h and then treated with sesamin (40 μM) for 24 h. A wound-healing assay was then performed, and we found that cell motility was further inhibited upon combinatorial treatment with MAPK inhibitors and sesamin, in comparison with sesamin treatment alone, irrespective of the treatment duration (Figure 7a,b). Furthermore, in FaDu cells, combinatorial treatment with the p38 and p-JNK inhibitors reduced cell motility by 26% and 42% relative to that resulting from sesamin treatment alone (Figure 7c,d). In addition, HSC-3 and FaDu cells were pretreated with SB203580 and SP600125 for 1 h and then with sesamin (40 μM) for 24 h, and the cells were seeded in the upper chamber of the transwell apparatus. Consequently, combinatorial treatment with SB203580 and SP600125 along with sesamin significantly reduced the migration of cells in comparison with sesamin treatment alone in both cell lines (Figure 8a). The quantitative findings are shown in Figure 8b,c. These results indicate that sesamin potentially reduces the migration and motility of oral cancer cells by suppressing p38 and JNK phosphorylation.

## 4. Discussion

Tumorigenesis is a complex phenomenon. According to the hallmarks of cancer [25], the mechanism underlying cancer cell migration and invasion has been assessed in numerous studies to determine the carcinogenicity of substances. Cancer cells secrete numerous proteases, including MMPs, urokinase plasminogen, and cathepsin, to degrade the basement membrane and extracellular matrix, thus facilitating the passage of cancer cells into the circulation to metastasize to distant organs [26]. Furthermore, MMP-2 is associated with metastasis in oral cancer, nasopharyngeal carcinoma, eye cancer, and brain cancer [23,27,28,29]. Therefore, we expected that MMP-2 inhibition would suppress metastasis. Hence, we examined the metastatic properties of the oral cancer cell lines HSC-3, FaDu, and Ca9-22 upon sesamin treatment. As expected, our results indicated that sesamin can reduce MMP-2 protein expression and further inhibit the migration and invasion ability of three oral cancer cell lines.

Sesame seeds are rich in oil, tiny, and considered healthy for consumption in Asia to increase daily energy intake. Studies have reported that sesame is rich in α-tocopherol, which can increase γ-tocopherol levels in the plasma and promote the bioactivity of vitamin E to delay aging [30]. The high levels of unsaturated fatty acids and magnesium in sesame oil are also beneficial for reducing blood pressure and plasma LDL-cholesterol levels [31,32]. Furthermore, one study reported that the low carbohydrate content of sesame aids blood sugar control and mitigates insulin resistance in hepatic L02 cells [33], which suggests that sesame may alleviate metabolic syndrome, thus preventing cancer metastasis [34]. Sufficient plasticity and energy can be achieved with a healthy diet and lifestyle and strongly influence tumor progression. Some studies have reported that sesamin inhibits DNA synthesis and affects cell cycle progression from the G_1_ phase in breast cancer cells [35]. In cervical cancer, sesamin induces ER stress-mediated apoptosis and inhibits proliferation/migration in HeLa cells [36]. The present results show that that sesamin inhibited the motility, migration, and invasion of and downregulated MMP-2 in oral cancer cells without exerting any cytotoxic effects.

To better clarify the association of sesamin upstream or downstream of MMP-2 signaling, we assessed the expression of MAPK pathway proteins, which are frequently involved in metastasis, including ERK1/2, p38, and JNK1/2 [37,38,39]. The present results show that sesamin reduced the phosphorylation of p38 and JNK1/2 but increased the phosphorylation of ERK1/2 in both HSC-3 and FaDu cells. Peng et al. (39) reported that the reduction of p38 and JNK1/2 phosphorylation in oral cancer cells further inhibits MMP-2 expression and cell migration but does not affect ERK1/2 phosphorylation. The present study reveals that sesamin increases ERK1/2 phosphorylation in a dose-dependent manner. One study reported that in human glioblastoma multiforme cell lines, tricetin treatment (80 μM) suppressed cell migration/invasion by enhancing ERK1/2 phosphorylation and inhibited MMP-2 expression [40]. The treatment of cells with a specific ERK inhibitor further confirmed that MMP-2–mediated cell migration can be well controlled [40]. Together, these effects may have been observed because of the cellular protective effects against exogenous drugs [40]. ERK activation prevents cancer cell apoptosis [41]. Furthermore, the activation of the ERK/MAPK pathways is also involved in the regulation of LC3 and cathepsin S proteins and is potentially associated with cell autophagy and metastasis in oral cancer cells [42]. Therefore, MAPK proteins are implicated in numerous processes in cancer progression and are therefore worthy of further investigation.

## 5. Conclusions

In conclusion, this study shows that sesamin has antimetastatic effects in HNSCC through the targeting of MMP-2 and reduction of p38 and JNK1/2 phosphorylation. The present results suggest that sesamin is not only a potential energy supplement for humans but also serves as a potential chemotherapeutic drug to treat metastatic oral cancer.

## Figures and Tables

**Figure 1 molecules-25-02248-f001:**
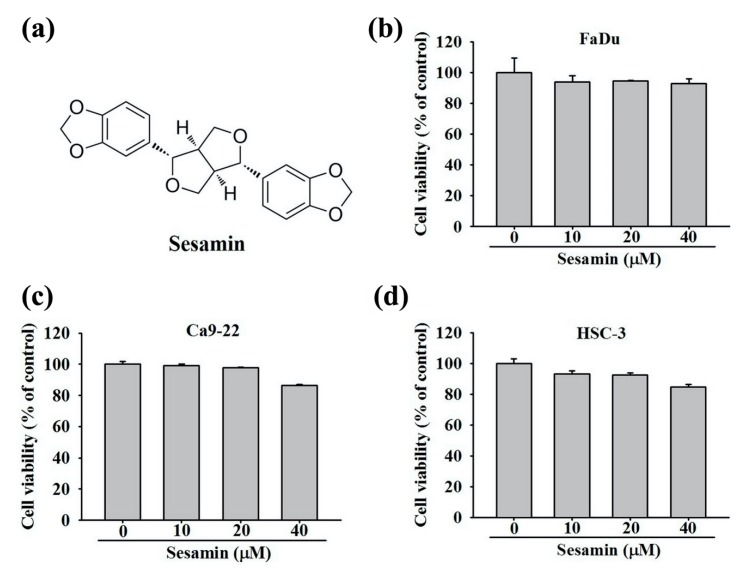
Cell viability of human oral cancer cells following sesamin treatment. Human oral cancer cell lines FaDu, Ca9-22, and HSC-3 were treated with sesamin (0, 10, 20, and 40 μM) for 24 h and then subjected to MTT assays for cell viability analysis (**a**) Chemical structure of sesamin (SES). (**b**) FaDu, (**c**) Ca9-22, and (**d**) HSC-3 cell viabilities displayed as percentages. The values represent the means ± SD of at least three independent experiments.

**Figure 2 molecules-25-02248-f002:**
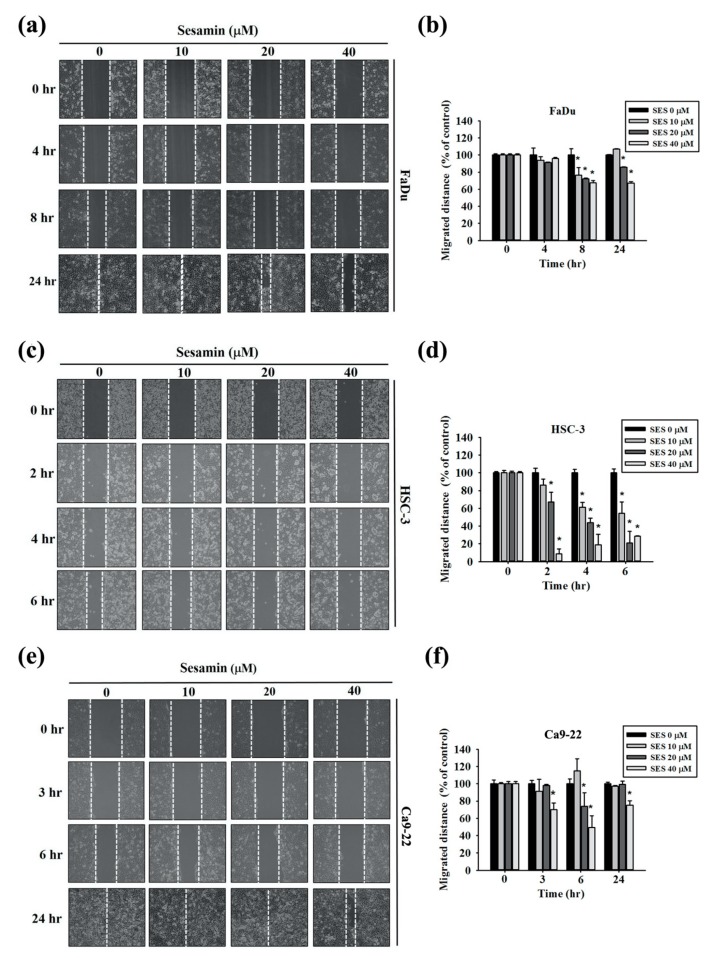
Cell motility according to the wound-healing assay of human oral cancer cells following sesamin treatment. (**a**) FaDu cells were seeded into 6-well plates in appropriate numbers. Photographs show wound closure following treatment with sesamin (SES) (0, 10, 20, or 40 μM) for 4 and 8 h. (**b**) The quantitative analysis shows the FaDu cell movement distance. (**c**) HSC-3 cells shown at the different time points of 2, 4, and 8 h by wound closure photographs and (**d**) the quantitative results. (**e**,**f**) Ca9-22 cells shown at 3, 6, 24 h by wound closure photographs. The values represent the means ± SD of at least three independent experiments. * *p* < 0.05, compared with the control group.

**Figure 3 molecules-25-02248-f003:**
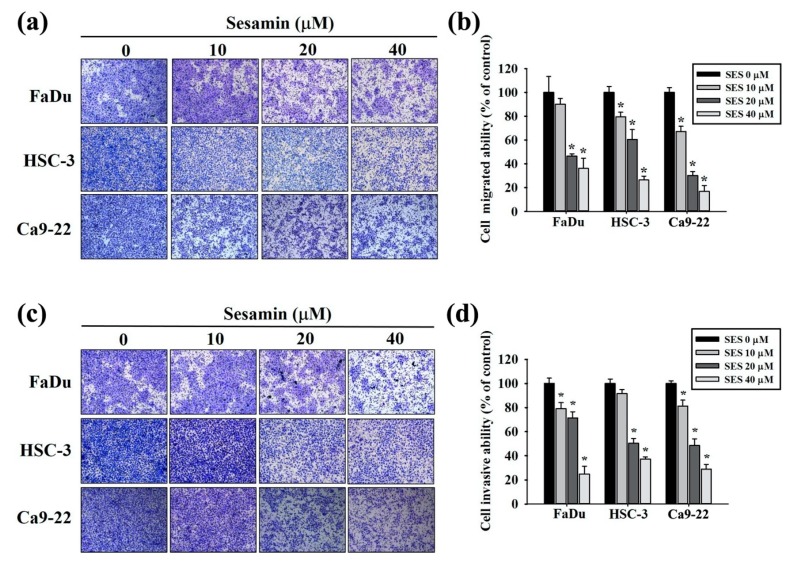
Cell invasion and migration of human oral cancer cells following sesamin treatment. (**a**) FaDu, HSC-3, and Ca9-22 cell lines were treated with SES (0, 10, 20, and 40 μM) in serum-free medium for 24-h and subjected to transwell assays as described in the “Material and Methods” section. (**b**) Quantitative results for the FaDu, HSC-3, and Ca9-22 cell lines; the numbers of cells that migrated to the underside of the porous polycarbonate. (**c**) FaDu, HSC-3, and Ca9-22 cell lines were treated with SES (0, 10, 20, and 40 μM); cell invasion was subsequently measured using Matrigel-coated transwell inserts as described in the “Material and Methods” section. (**d**) Quantitative results for the FaDu, HSC-3, and Ca9-22 cell lines; the numbers of cells that invaded the underside of the porous polycarbonate. The values represent the means ± SD of at least three independent experiments. * *p* < 0.05, compared with the control group.

**Figure 4 molecules-25-02248-f004:**
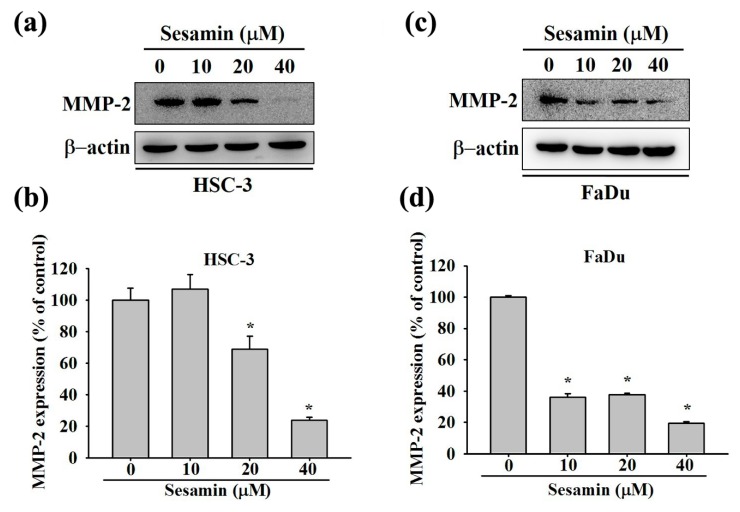
MMP-2 protein expression in human oral cancer cells following sesamin treatment. (**a**) HSC-3 and (**c**) FaDu cells were treated with different concentrations of SES (0, 10, 20, and 40 μM), and the cell lysates were subjected to SDS-PAGE followed by Western blotting as described in Materials and Methods. (**b**,**d**) The quantitative results for the HSC-3 and FaDu cells, individually. MMP-2 protein expression after being normalized to β-actin. The values represent the means ± SD of at least three independent experiments. * *p* < 0.05, compared with the control group.

**Figure 5 molecules-25-02248-f005:**
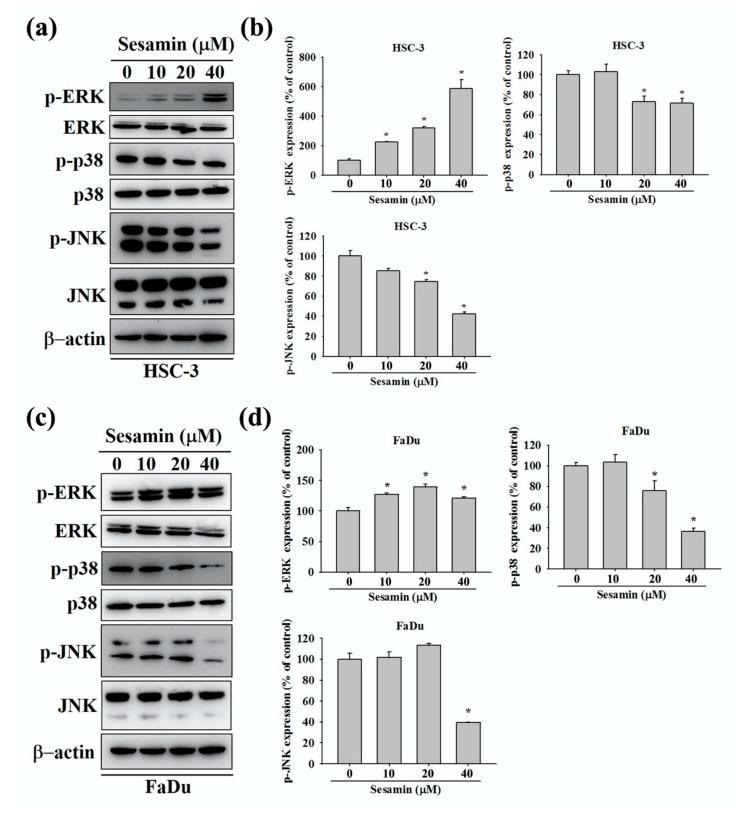
Mitogen-activated protein kinase activation in human oral cancer cells following sesamin treatment. (**a**) HSC-3 and (**c**) FaDu cells were treated with different concentration of SES (0, 10, 20, and 40 μM), and the cell lysates were subjected to SDS-PAGE followed by Western blotting to detect the antibodies for phosphorylated ERK 1/2, p38, and JNK1/2. (**b**,**d**) show the quantitative results for the expression of each protein, after being adjusted to β-actin, in HSC-3 and FaDu cells. The values represent the means ± SD of at least three independent experiments. * *p* < 0.05, compared with the control group.

**Figure 6 molecules-25-02248-f006:**
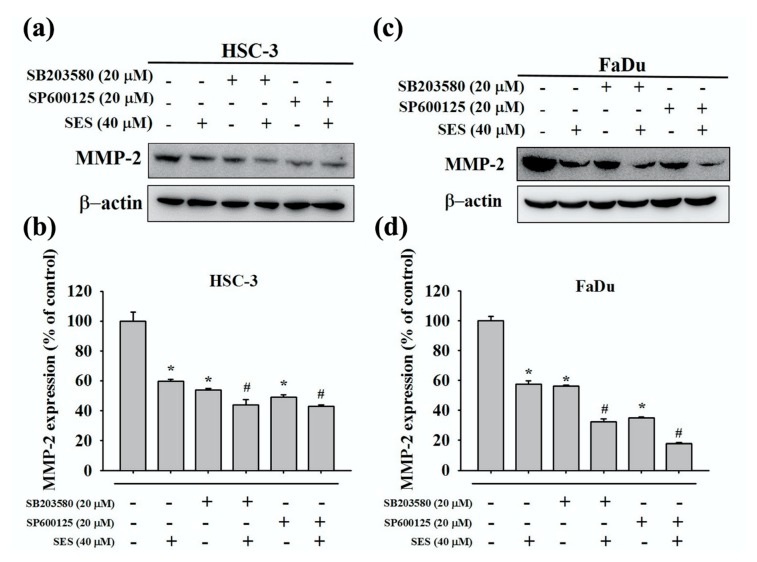
The relationship between MMP-2 and MAPK inhibitors in human oral cancer cells following sesamin treatment. (**a**) HSC-3 and (**c**) FaDu cells were pretreated with SP600125 or SB203580 for 1 h and then incubated in the presence or absence of 40 μM SES for 24 h. After that, Western blot assays were performed to detect the antibody for MMP-2. (**b**,**d**) show the quantitative results of the expression of each protein, after being adjusted to β-actin, in HSC-3 and FaDu cells. The values represent the means ± SD of at least three independent experiments. * *p* < 0.05, compared with the control group. # *p* < 0.05, compared with the SES-treated group.

**Figure 7 molecules-25-02248-f007:**
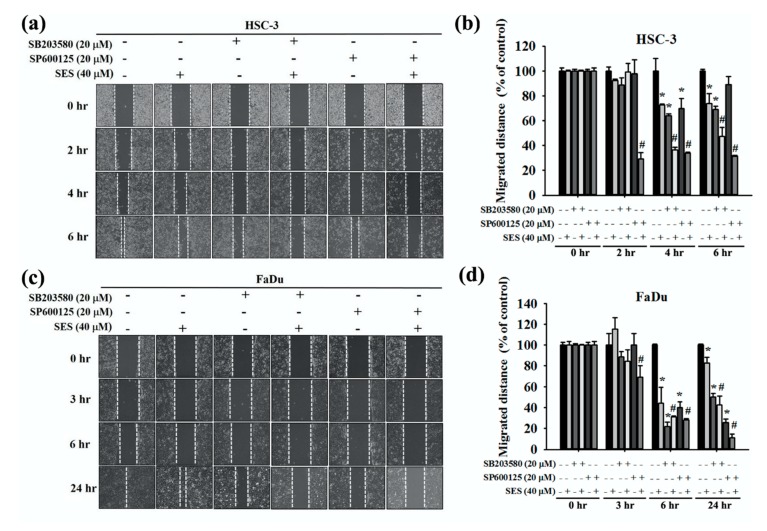
The relationship between cell motility and MAPK inhibitors in human oral cancer cells following sesamin treatment. (**a**) HSC-3 and (**c**) FaDu cells were pretreated with SP600125 or SB203580 for 1 h and then incubated in the presence or absence of 40 μM SES for 24 -h. After that, wound-healing assays were performed at different time points. (**b**) HSC-3 cells shown at 2, 4, and 6 h, depicting the wound closure distances for quantitative analysis. (**d**) FaDu cells shown at 3, 6, and 24 h, depicting the wound closure distances for quantitative analysis. The values represent the means ± SD of at least three independent experiments. * *p* < 0.05, compared with the control group. # *p* < 0.05, compared with the SES-treated group.

**Figure 8 molecules-25-02248-f008:**
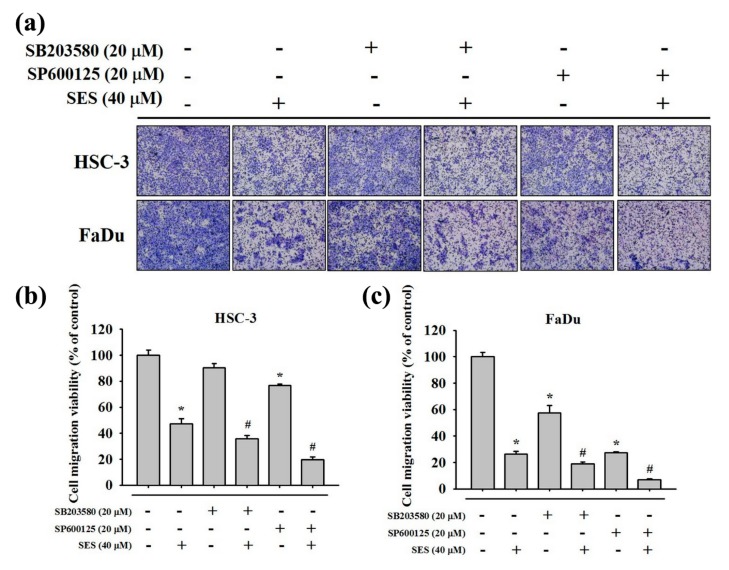
The relationship between cell migration and MAPK inhibitors in human oral cancer cells following sesamin treatment. (**a**) HSC-3 and FaDu cells were pretreated with SP600125 or SB203580 for 1 h and then incubated in the presence or absence of 40 μM SES for 24 h. After that, transwell assays were performed as described in the “Material and Methods” section. (**b**) Quantitative results for the HSC-3 cell line, determining the number of cells that migrated to the underside of the porous polycarbonate. (**c**) Quantitative results for the FaDu cell line, determining the number of cells that migrated to the underside of the porous polycarbonate. The values represent the means ± SD of at least three independent experiments. * *p* < 0.05, compared with the control group. # *p* < 0.05, compared with the SES-treated group.

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
