# Peer review of "Antimetastatic Effects of Sesamin on Human Head and Neck Squamous Cell Carcinoma through Regulation of Matrix Metalloproteinase-2"

_molecules, 2020, doi:10.3390/molecules25092248_

Round 1

Reviewer 1 Report

The natural-derived compounds, such as sesamin in the present work, in in vitro and in vivo experiments, are the reseach area of many gropus, as these compounds usualy present less toxicity than the synthesized drugs. This manuscript is describing the potential antimetastatic effects of sesamin on human head and neck squamous cell carcinoma.

The authors reported on the evaluation of the cell viability of human oral cancer (HSC-3, FaDu and Ca9-22 cells) by using MTT assay, then a wound-healing assay was performed for FaDu, HSC-3, and Ca9-22 cells with 0, 10, 20, and 40 μM sesamin in order to assess the coordinated movement of a cell population. The results indicate that sesamin is not toxic and inhibits the coordinated movement of the tumor cell population. Sesamin inhibited cell migration in a dose-dependent manner in all 3 cell lines, but is effective at the highest concentration used, 40 μM, as it was revelead by a transwell assay.

Also, it was shown by using western blotting that sesamin downregulates MMP-2 in HSC-3 and FaDu cells, and sesamin cooperatively downregulates MMP-2, p-JNK1/2, and P-p38 and inhibits the MAPK signaling pathway along with p38 and p-JNK inhibitors in oral cancer cells (HSC-3 and FaDu cells). The wound healing assay demonstrated that sesamin potentially reduces the migration and motility of oral cancer cells by suppressing p38 and JNK phosphorylation.

The data presented are strong, and convincingly exhibited the antimetastatic effects of sesamin in in vitro models. The manuscript is concise and the appropriate analyses are performed.

This is a well performed study that I consider that is important and represent a new strategy to identify and evaluate new natural-derived molecules with good antimetastatic activities.

The authors need to address the below comments to strengthen quality of the manuscript:

Please indicate the number of cells used in each experiments, as usually the results are also dependent on the number of cells used (e.g. cell viability assay).

Line 91: “Sesamin is a pure substance bought from a company called Sigma-Aldrich (St. Louis, MO, USA)”. Replace by “Sesamin is a pure substance bought from Sigma-Aldrich (St. Louis, MO, USA)”.

Line 96: „Cells were put into a 37 degrees incubator following the required conditions as well as time of the experiment.”  Replace by„Cells were put into an incubator 37 °C and 5% CO2 following the required conditions as well as time of the experiment.” Please add all over the text „37 °C” and not ‚37 degrees”.

Line 107: Please rewrite the phrases to sound more scientifically correct : „The absorption of the solution under 595 nm by utilizing the spectrophotometer, and the number of the living cells could be known through absorptivity.” „Albuminimetry adopted the means of Bradford protein assay, drew a standard curve with 570nm wavelength visible light absorption, tested the sample’s absorption in the same way, and the concentration of the sample’s protein could be calculated based on the standard curve.”

Please write the model of the equipment used in spectrophotometric measurements. 

Please check the misspelling in the text.

Author Response

The natural-derived compounds, such as sesamin in the present work, in in vitro and in vivo experiments, are the reseach area of many gropus, as these compounds usualy present less toxicity than the synthesized drugs. This manuscript is describing the potential antimetastatic effects of sesamin on human head and neck squamous cell carcinoma.

The authors reported on the evaluation of the cell viability of human oral cancer (HSC-3, FaDu and Ca9-22 cells) by using MTT assay, then a wound-healing assay was performed for FaDu, HSC-3, and Ca9-22 cells with 0, 10, 20, and 40 μM sesamin in order to assess the coordinated movement of a cell population. The results indicate that sesamin is not toxic and inhibits the coordinated movement of the tumor cell population. Sesamin inhibited cell migration in a dose-dependent manner in all 3 cell lines, but is effective at the highest concentration used, 40 μM, as it was revelead by a transwell assay.

Also, it was shown by using western blotting that sesamin downregulates MMP-2 in HSC-3 and FaDu cells, and sesamin cooperatively downregulates MMP-2, p-JNK1/2, and P-p38 and inhibits the MAPK signaling pathway along with p38 and p-JNK inhibitors in oral cancer cells (HSC-3 and FaDu cells). The wound healing assay demonstrated that sesamin potentially reduces the migration and motility of oral cancer cells by suppressing p38 and JNK phosphorylation.

The data presented are strong, and convincingly exhibited the antimetastatic effects of sesamin in in vitro models. The manuscript is concise and the appropriate analyses are performed.

This is a well performed study that I consider that is important and represent a new strategy to identify and evaluate new natural-derived molecules with good antimetastatic activities.

The authors need to address the below comments to strengthen quality of the manuscript:

Please indicate the number of cells used in each experiment, as usually the results are also dependent on the number of cells used (e.g. cell viability assay).

Answers: Thanks for this valuable suggestion. The follow sentence has been modified in revised manuscript.

Line 98: HSC-3, FaDu and Ca9-22 cells were evenly distributed into 96-well plates with appropriate number of cells (1 x 104 cells/well).

Line 110: HSC-3, FaDu and Ca9-22 oral cancer cells were evenly distributed into culture-inserts 2 Well for self-insertion on 6-well plates (2 x 105 cells/well) and incubators with appropriate number of cells and given different concentrations of sesamin (0, 10, 20, 40 μM).

Line 116: HSC-3, FaDu and Ca9-22 oral cancer cells were evenly distributed into 6 cm petri dishes with appropriate number of cells (4 x 105 cells/well), and placed at 37°C, 5% CO2 incubator for 24-h-cultivation. After removing the used culture fluid, sesamin (0, 10, 20, 40 μM) was added for 37 °C, 24-h-cultivation. By making use of the transwell (1 x 104 cells/well).

Line 127: HSC-3, FaDu and Ca9-22 oral cancer cells were evenly distributed into 6 cm petri dishes with appropriate number of cells (4 x 105 cells/well).

Line 91: “Sesamin is a pure substance bought from a company called Sigma-Aldrich (St. Louis, MO, USA)”. Replace by “Sesamin is a pure substance bought from Sigma-Aldrich (St. Louis, MO, USA)”.

Answers: Thanks for this valuable suggestion. The follow sentence has been modified in revised manuscript.

Line 91: The follow sentence “Sesamin is a pure substance bought from a company called Sigma-Aldrich (St. Louis, MO, USA)” has been modified to “Sesamin is a pure substance bought from Sigma-Aldrich (St. Louis, MO, USA)” in revised manuscript.

Line 96: „Cells were put into a 37 degrees incubator following the required conditions as well as time of the experiment.”  Replace by „Cells were put into an incubator 37 °C and 5% CO2following the required conditions as well as time of the experiment.” Please add all over the text „37 °C” and not ‚37 degrees”.

Answers: Thanks for this valuable suggestion. The follow sentence has been modified in revised manuscript.

Line 96: The follow sentence “Cells were put into a 37 degrees incubator following the required conditions as well as time of the experiment.” has been modified to “Cells were put into an incubator 37°C and 5% CO2 following the required conditions as well as time of the experiment.” in revised manuscript.

Line 107: Please rewrite the phrases to sound more scientifically correct: „The absorption of the solution under 595 nm by utilizing the spectrophotometer, and the number of the living cells could be known through absorptivity.” „Albuminimetry adopted the means of Bradford protein assay, drew a standard curve with 570nm wavelength visible light absorption, tested the sample’s absorption in the same way, and the concentration of the sample’s protein could be calculated based on the standard curve.”

Answers: Thanks for this valuable suggestion. The follow sentence has been modified in revised manuscript.

Line 107: The follow sentence “The absorption of the solution under 595 nm by utilizing the spectrophotometer, and the number of the living cells could be known through absorptivity.” has been modified to “The crystals were dissolved with methanol and absorbance was determined at 595 nm by utilizing the spectrophotometer (BioTek Synergy HTX Multi-Mode Reader). Each concentration will be repeated in three separate experiments.” in revised manuscript.

Line 130: The follow sentence “Albuminimetry adopted the means of Bradford protein assay, drew a standard curve with 570nm wavelength visible light absorption, tested the sample’s absorption in the same way, and the concentration of the sample’s protein could be calculated based on the standard curve.” has been modified to “The protein concentration of total cell lysates was determined by BCA protein assay. The absorbance was determined at 570 nm by utilizing the spectrophotometer (BioTek Synergy HTX Multi-Mode Reader).” in revised manuscript.

Please write the model of the equipment used in spectrophotometric measurements. 

Answers: Thanks for this valuable suggestion. The model of spectrophotometric measurements has been added in revised manuscript.

Line 107: The follow sentence “The crystals were dissolved with methanol and absorbance was determined at 595 nm by utilizing the spectrophotometer (BioTek Synergy HTX Multi-Mode Reader). Each concentration will be repeated in three separate experiments.” has been modified in revised manuscript.

Please check the misspelling in the text.

Answer: We apologize for these confuse. The misspelling has been proofed and highlight with red in revised manuscript.

Reviewer 2 Report

The manuscript entitled "Antimetastatic effects of sesamin on human head and 3 neck squamous cell carcinoma through regulation of 4 matrix metalloproteinase‐2" is describing the role of sesamin in tumor metastasis.

Authors are presenting clear results and the findings described here is very interesting. I think that this manuscript should be accepete for publication.

Author Response

The manuscript entitled "Antimetastatic effects of sesamin on human head and neck squamous cell carcinoma through regulation of matrix metalloproteinase‐2" is describing the role of sesamin in tumor metastasis.

Authors are presenting clear results and the findings described here is very interesting. I think that this manuscript should be accepted for publication.

Answers: Thanks for this valuable suggestion.
